# Optical Coherence Tomography Findings in Rhegmatogenous Retinal Detachment: A Systematic Review

**DOI:** 10.3390/jcm11195819

**Published:** 2022-09-30

**Authors:** Carla Danese, Paolo Lanzetta

**Affiliations:** 1Department of Medicine—Ophthalmology, University of Udine, 33100 Udine, Italy; 2Istituto Europeo di Microchirurgia Oculare (IEMO), 33100 Udine, Italy

**Keywords:** OCT, optical coherence tomography, retinal detachment, rhegmatogenous retinal detachment, angiography, en face, adaptive optics, biomarkers

## Abstract

Rhegmatogenous retinal detachment is a sight-threatening condition that may lead to blindness if left untreated. Surgical treatments may vary and are tailored to a single patient. Anatomical and functional results may vary, due to factors that are currently under study. Optical coherence tomography (OCT) allows a detailed visualization of the retinal structure. Some studies have been performed using OCT on eyes with retinal detachment. We performed a review on the subject. Several data have been obtained using different OCT applications. Some alterations may represent potential biomarkers since they are associated with visual and anatomical prognoses. Increased knowledge on the subject may be helpful to choose among different surgical strategies and endotamponades. More research on the topic is needed.

## 1. Introduction

Retinal detachment is defined as the separation of the neurosensory retina from the underlying retinal pigment epithelium (RPE). In its rhegmatogenous form, liquefied vitreous detaches the retina by passing through a retinal tear or hole. Its incidence is around 1 in 10,000 persons per year [1]. According to the macular involvement, it may be classified as “macula on” or “macula off” retinal detachment. Especially when it is of recent onset and without macular involvement, it is considered a surgical emergency. Potential risks of delayed surgery are a progression of the detached retina, development of proliferative vitreoretinopathy, and worse functional outcomes if the macula becomes involved [2]. It may lead to blindness of the affected eye unless surgical treatment is promptly performed. There is no ideal strategy for its treatment. The currently available options are scleral buckling, pars plana vitrectomy, and pneumatic retinopexy. Each one has characteristic advantages for certain patients. After vitrectomy, a tamponade with gas or silicone oil is required [3]. In some cases, visual results may be unsatisfying. In order to better understand factors affecting postoperative prognosis, several studies have been conducted correlating optical coherence tomography (OCT) findings with clinical outcomes [4]. Spectral-domain OCT (SD-OCT) and swept-source OCT (SS-OCT) are both types of Fourier domain OCT. Nowadays, they have a high sensitivity, providing high imaging speed, improved image contrast, full volumetric tissue information, and high resolution in the three dimensions [5]. OCT usually refers to B-scan imaging, derived from sagittal and transverse sections. C-scan, or en face OCT, is an application of SD-OCT, producing frontal sections of retinal layers [6]. Three-dimensional images are obtained with serial horizontal or vertical B-scan images, which are reconstructed in a three-dimensional “cube” [7]. OCT angiography (OCT-A) is a novel development, allowing visualization of the retinal vascularization without a contrast agent [5]. Recently, instruments using adaptive optics have been combined with SS-OCT imaging (AO-OCT). Adaptive optics produce a two-dimensional areal image of the retina with a cellular resolution. Therefore, this multimodal retina imaging system may add cellular resolution to images obtained with SS-OCT [8].

This review aims to be an update of the knowledge on the subject so far, taking into account SD-OCT, SS-OCT, OCT-A, three-dimensional OCT (3D-OCT), and AO-OCT.

Although some authors have performed OCT studies specifically focused on the postoperative period in order to study surgical complications, such as macular hole formation, these findings will be only briefly touched on the present review [9,10].

## 2. Materials and Methods

Articles published in PubMed, without restriction on the year of publication and published until July 2022, were considered. Appropriate keywords were used in order to retrieve research articles on rhegmatogenous retinal detachment and OCT. Only articles in English wereincluded. Research works conducted with time-domain OCT were not included.

Table 1, Table 2, Table 3, Table 4 and Table 5 report the included publications, divided according to the type of study and OCT examination.

## 3. Results

### 3.1. Spectral-Domain OCT and Swept-Source OCT

Studies conducted on macula-off retinal detachment showed that the baseline visual acuity and time to surgical repair are among the best predictors of vision outcomes [11,12]. The number of detached quadrants and the stage of proliferative vitreoretinopathy are also related to poor postoperative anatomical results and poor visual prognosis [11,12,13]. In addition, the integrity of the ellipsoid zone and the external limiting membrane is useful to predict postoperative visual acuity [12,14,15,16,17]. However, Sridhar and colleagues reported that two patients with macula-off retinal detachment experienced a postoperative improvement in visual acuity secondary to ellipsoid zone restoration [18]. Changes in the thickness of the outer nuclear layer may also predict the postoperative visual outcome [19]. A lower mean preoperative central retinal thickness is associated with a good visual prognosis [20].

Terauchi and colleagues found that the thickness of the inner segments of the photoreceptors was significantly thinner in the early postoperative period. They also showed that the thickness of the inner and outer segments of the photoreceptors in an early postoperative period may be a good indicator of the final visual acuity [21].

Some authors found a correlation between outer retinal folds and postoperative metamorphopsia. They also showed abrupt changes in the reflectivity of the ellipsoid zone, associated with folds [22]. Nagpal and colleagues found that outer retinal corrugation was associated with a poor postoperative visual outcome. Therefore, it may be an important predictor of visual outcome [17].

Poulsen and colleagues demonstrated that in eyes with macula-off retinal detachment, a detached macula with a near-normal appearance had a better visual prognosis than a detached macula with a disrupted intraretinal appearance [23].

Some authors observed that the percentage of eyes with integrity of the photoreceptor layer increased progressively over time after surgery for macula-off retinal detachment. A delayed surgery was associated with a higher risk of layer disruption, and it was therefore associated with a worse visual prognosis. Cystoid macular edema and epiretinal membranes were associated with a lower postoperative visual acuity [24].

Iwase and colleagues studied intraretinal cystoid cavities in macula-off retinal detachment. They found that they were associated with the anterior protrusion of the macula. While their presence was associated with worse preoperative visual function and morphology, it did not affect postoperative outcomes [25].

OCT allows the recognition of whitish outer retinal spots, occasionally appearing in the detached retina as hyperreflective foci in the ellipsoid or interdigitation layers [26].

SD-OCT permits the detection of microscopic macular changes in macula-off retinal detachment. Zgolli and colleagues measured the height of the subretinal fluid, finding that a greater level of macular detachment was correlated with lower preoperative and postoperative visual acuity and with the formation of cavitations in the external nuclear layer [13]. Other authors, on the other hand, found no statistically significant correlation between the height of macula-off retinal detachment and the final visual acuity. They observed that the presence of a macular hole was the only preoperative variable with a significant correlation with postoperative visual acuity [27].

Outer retinal undulation is a debated potential biomarker. It is thought to be caused by a disparity in the amount of edema between the inner and outer retina. This is likely caused by more severe damage in the outer retina than in the inner retina. Yeo and colleagues did not find a significant influence of outer retinal undulation on visual outcomes in patients with macula-off retinal detachment. Performing SS-OCT, they also showed that outer retinal undulation was associated with younger age and better preoperative visual acuity. Most importantly, patients with a recent retinal detachment had a higher incidence of outer retinal undulation, which may therefore be used to determine retinal detachment duration in patients with an unknown duration of symptoms [14].

Research has been conducted in order to identify factors associated with postoperative metamorphopsia following successful vitrectomy for retinal detachment. Using SS-OCT, Kumar and colleagues found that the preoperative extent of the detachment, postoperative foveal contour, and the continuity of the ellipsoid zone are significantly associated with the occurrence of postoperative metamorphopsia [28].

Mané et al. conducted a retrospective study using SD-OCT. They observed that preoperative OCT examination detected a shallow macular detachment extending beyond the fovea in patients diagnosed with fovea-splitting retinal detachment by clinical examination with a non-contact lens. These patients were considered to have macula-off from an anatomical point of view, but their postoperative prognosis was closer to macula-on. The authors assumed that the favorable postoperative functional outcome in these eyes was due to a shallow macular detachment with moderate preoperative visual loss and a short duration of the detachment, allowing the restoration of the ellipsoid zone and external limiting membrane [29].

Klaas and colleagues used SD-OCT to define a more precise classification of rhegmatogenous retinal detachment according to the state of the macula and the fovea. They also showed that the grade of retinal detachment and the extent of the cystoid macular edema were good preoperative biomarkers to predict functional recovery in eyes with detached fovea [30].

Bansal et al. monitored retinal reattachment with SS-OCT after pneumatic retinopexy, identifying five specific stages. Stage 1 is characterized by a rapid reduction in the height of the detachment, leading to an improvement in the metabolic transport between the retina and the RPE and dehydration of the inner and outer retina. The subsequent reduction in cystoid macular edema and the improvement in hydration folds and outer retinal corrugations define stage 2. In stage 3, the retina makes contact with the RPE. The subsequent rapid deturgescence of the inner and the outer segments of the photoreceptors defines stage 4. Stage 5 is characterized by an improvement in the integrity of the outer retina, involving the external limiting membrane and the ellipsoid zone, and eventual recovery of the foveal bulge. The authors also observed that in eyes with acute retinal detachment and symptom occurrence lasting less than 24 h, the outer retina may still be intact, therefore achieving a quicker recovery of the foveal bulge [31].

When performing SD-OCT on eyes that underwent a pars plana vitrectomy for macula-off retinal detachment, Ozsaygili and colleagues observed that different endotamponades may have different effects on the thickness of the retinal layers. No significant difference was observed in the eyes after gas tamponade. On the other hand, significant changes, especially in the ganglion cells layer and in the outer nuclear layer, were seen in the eyes when silicone oil was used; they also had a worse visual recovery. The difference in the thickness of the ganglion cells layer showed the strongest correlation with worse visual outcomes [32]. These results confirmed a similar finding from other authors, who found that a reduction in the thickness of the ganglion cell layer and inner plexiform layer after tamponade with silicone oil, with an unclear mechanism [33]. The thickness of the ganglion cells layer and inner plexiform layer may be a predictive factor for the final visual acuity, according to Raczynska and colleagues [34]. Lee et al. found that silicone oil tamponade may cause a temporary thinning of the parafoveal inner retina, which recovered after silicone oil removal. However, the thinning of the peripapillary nerve fiber layer remained unchanged after removal. These changes were likely due to the mechanical pressure of the endotamponade on the retina [35].

Horozoglu et al. found that the long-term use of heavy silicone oil for the treatment of retinal detachment resulted in a good anatomical reattachment of the retina with good ellipsoid zone continuity and foveal thickness. However, extended use of heavy silicone oil increased the rate of epiretinal membrane formation [36].

There is still no consensus regarding the best surgical strategy to treat retinal detachment. The advantages of one technique over the other are still debated. Although OCT studies may provide useful information on the most indicated surgical approach, there is still limited evidence to draw any substantial conclusion [37].

It has been suggested that postoperative discontinuity of the ellipsoid zone and of the external limiting membrane and the development of outer retinal folds, may be influenced by the surgical technique [38,39]. Some authors found that outer retinal folds were associated with significantly worse visual acuity. In addition, there was a negative correlation between the closest distance of an outer retinal fold from the fovea and vertical metamorphopsia [39].

Stopa and colleagues found that the functional outcome of eyes affected by retinal detachment complicated by proliferative vitreoretinopathy might be influenced by an abnormal macular status, which can be found in the majority of these eyes [40].

The persistent subfoveolar fluid following surgery for macula-off retinal detachment has been studied by Tee et al. They found that the fluid was almost always present in eyes with retinal detachment secondary to atrophic round holes or dialyses, while it was present in one third of eyes with a detachment secondary to tractional tears. The fluid may persist for up to one year. It slowly resolves with time, but in some cases it is associated with the development of progressive foveal photoreceptor atrophy and loss of visual acuity [41].

Borowicz and colleagues found that macular changes shown through OCT examination occurred postoperatively both in macula-on and macula-off eyes. Specifically, they identified epiretinal membranes, macular edema, subretinal fluid, and increased central retinal thickness [42].

### 3.2. OCT Angiography

Some authors have performed OCT-A after rhegmatogenous retinal detachment in order to investigate visual prognostic factors with vessel density (VD) measurements. Hong and colleagues found that the postoperative state of the outer retinal layer was associated with the subfoveal choriocapillaris VD, which correlated well with visual outcomes. An intact outer retina was associated with a normal VD of the choriocapillaris, while patients with outer retinal defects presented a significantly lower choriocapillaris VD [43].

After successful pars plana vitrectomy for macula-off retinal detachment, Chatziralli et al. found an enlargement of the foveal avascular zone (FAZ), accompanied by a significant decrease in VD in both superficial and deep capillary plexuses. A significant thinning of inner retinal layers was also observed, corresponding to the areas of decreased VD [44].

In a study by Xu and colleagues, the superficial FAZ was found to be normal, while the deep FAZ was enlarged postoperatively. Moreover, in eyes with rhegmatogenous retinal detachment and choroidal detachment, the deep FAZ continued to increase in size in the postoperative period. In this group of patients, there was a significant negative correlation between the deep FAZ area and visual acuity [45].

Some authors confirmed that the FAZ area increased and VD decreased when compared to normal eyes. The parafoveal VD progressively increased in the postoperative period, but it did not reach a normal status. However, these alterations did not have a correlation with visual outcomes. They also found that eyes with a thinner preoperative foveal sensory thickness presented a lower VD in the superficial capillary plexus postoperatively. The VD of the superficial capillary plexus was lower in eyes treated with pars plana vitrectomy than in eyes treated with scleral buckling [46].

Roohipoor and colleagues performed OCT-A on eyes with silicone oil tamponade following vitrectomy for macula-off retinal detachment. They found a significantly lower VD of the parafoveal superficial capillary layer and the total retina, compared to a normal eye. The parafoveal VD progressively increased postoperatively, but it did not return to normal. The VD of the deep capillary plexus and the choroidal flow were less than normal in silicone oil-filled eyes, but the difference did not reach statistical significance. In eyes with silicone oil tamponade, the FAZ was not affected [47]. Other authors found that the macular VD and FAZ were not affected by silicone oil [35].

### 3.3. En Face OCT

Some authors observed in some cases that en face OCT may represent a useful imaging technique in order to monitor outer retinal folds occurring postoperatively after vitrectomy. It is indeed well known that outer retinal folds are less likely to cause metamorphopsia and they spontaneously resolve after some months in the majority of the cases. En face OCT is useful since it may provide several pieces of structural information, especially on the outer retina [48,49].

En face OCT is also more sensitive than B-scan OCT for detecting epiretinal membrane formation after vitrectomy for the treatment of retinal detachment. Studies performed using en face OCT for detecting epiretinal membranes also showed that they have a marginal impact on postoperative visual acuity [50].

### 3.4. Three-Dimensional OCT

Hisatomi and colleagues used three-dimensional OCT (3D-OCT) to study the retinal changes after vitrectomy with the internal limiting membrane (ILM) peeling for the treatment of macula-on and macula-off rhegmatogenous retinal detachment. The observed changes were: thinning of the ILM peeling area, dissociation of the outer nerve fiber layer, dimple sign, temporal macular thinning, and forceps-related retinal thinning. These developed in the first two postoperative months and remained stable thereafter. Proliferative changes such as epiretinal membrane and proliferative vitreoretinopathy were not noted as a consequence of ILM peeling, suggesting that this maneuver may decrease the occurrence of this complication [51]. However, there is no extensive literature on the subject and this finding needs to be confirmed by further studies.

### 3.5. Adaptive Optics OCT

It has been suggested that post-surgical receptor regeneration may play a role in determining visual prognosis after retinal detachment. However, traditional OCT technology is not adequate to identify and track the evolution of single cone photoreceptors. Adaptive optics (AO) have been recently introduced to enhance ophthalmic imaging, analyzing single photoreceptors in vivo [4]. Some authors have tried to study photoreceptors using a fundus camera integrated with AO. In their studies, they found that cone density was reduced following retinal detachment, even if it improved after surgery. However, since fundus cameras provide two-dimensional images, information on all the retinal layers was captured, leading to artifacts and reduced visualization of single cones [52,53,54]. The combination of AO with OCT instead allows for increased lateral resolution, acquiring information for each individual retinal layer [55].

Reumueller et al. have conducted a prospective study on patients undergoing vitrectomy with gas tamponade for macula-off retinal detachment. They performed SD-OCT as well as AO-OCT with follow up times of 6 and 56 weeks after surgery. Even if cone morphology improved 6 weeks after surgery, significant distortion of the cone mosaic was still present after one year. This finding was correlated to reduced retinal sensitivity through microperimetry. Even if the visual acuity was satisfying, the regular mosaic of the cones appeared to be completely lost [4]. Distortion of cones may also explain the micrometamorphopsia reported by some patients despite macular reattachment after surgery in the absence of a secondary epiretinal membrane.

Table 6 summarizes possible biomarkers which have emerged from different studies.

## 4. Discussion

Rhegmatogenous retinal detachment is a potential cause of permanent vision loss. Its treatment requires surgical intervention, which aims to reattach the retina closing the breaks and releasing vitreoretinal tractions. Pneumatic retinopexy, scleral buckling, and vitrectomy are efficacious techniques with high success rates [56,57,58,59]. However, anatomical and functional postoperative success is affected by several factors.

OCT imaging enables the detection of structural changes which are not always evident in clinical examination. Therefore, several studies have been performed in order to obtain a better understanding of rhegmatogenous retinal detachment, and to identify possible biomarkers which may influence the prognosis [60,61].

In the literature, there are limited data on the subject. Most importantly, the majority of studies are retrospective in nature and only a few RCTs are available. Currently, a meta-analysis on the subject cannot be performed. The present narrative review sums up the different and sometimes conflicting evidence from the available studies to date. More research is needed in order to define the best practice pattern for the management of retinal detachment and its prognosis.

Actual knowledge on the physiology of retinal reattachment has improved thanks to studies using SS-OCT. This enhances the understanding of certain anatomic abnormalities occurring after retinal reattachments, such as outer retinal folds, outer retinal corrugations, and residual subfoveal fluid blebs [31].

Studies on OCT are not only useful to find new predictors of visual prognoses but also to improve the classification of rhegmatogenous retinal detachment. Traditionally, rhegmatogenous retinal detachment is classified according to the presence/absence of proliferative vitreoretinopathy and the status of the macula (macula-on/macula-off). In some cases of foveal-splitting retinal detachment diagnosed with clinical examination, SD-OCT allows the identification of a complete macular detachment, which was, however, associated with a better visual prognosis than a macula-off detachment. It may be useful to classify these cases as macula-on/off retinal detachment, in order to better stratify the visual prognosis [29].

Among the potential biomarkers predicting functional recovery, the grade of detachment and the extent of cystoid macular edema seems promising in eyes with a detached fovea, as well as the integrity of the ellipsoid zone [14,30]. The extent of the detachment, the postoperative foveal contour, the integrity of the ellipsoid zone and of the external limiting membrane, and the presence of outer retinal folds, seem to be predictors of visual prognoses and metamorphopsia following a successful vitrectomy [12,15,16,17,22,28,48,49]. In addition, the thickness of the inner and outer segments of the photoreceptors, and thickness changes in the outer nuclear layer, may be predictors of the final visual outcome [19,21]. Lower preoperative central retinal thickness is associated with a good visual prognosis [12,20]. Irregularity in the reflectivity of the ellipsoid zone may be associated with outer retinal folds. These alterations may represent subtle damage to the photoreceptors [22]. Outer retinal undulation may have a role in assessing the duration of retinal detachment, but it does not seem to be related to the visual prognosis [14]. Recovery of the integrity of the ellipsoid zone may be associated with postoperative improvement in visual acuity [18]. Persistent subfoveal fluid resolves spontaneously in the majority of cases, but it may also be rarely associated with progressive foveal photoreceptor atrophy and loss of visual acuity. It occurs more frequently when the macula-off retinal detachment is secondary to atrophic round holes or dialysis [41]. The quantity of macular subretinal fluid may also be considered a biomarker since it is correlated with low visual acuity and cavitations of the external nuclear layer [13]. On the other hand, other studies failed to find a significant correlation between the height of the retinal detachment and the visual outcome. The preoperative presence of a macular hole, instead, significantly affects postoperative visual acuity [27]. Intraretinal cystoid cavities do not seem to impair postoperative anatomical and functional outcomes [25].

OCT-A studies may also help to identify biomarkers useful for predicting functional and anatomical visual prognoses. OCT-A findings have suggested a potential relationship between outer retinal restoration, the VD of the choriocapillaris, and visual prognosis in macula-off retinal detachment after a vitrectomy [43]. In these patients, an enlargement of the FAZ with reduced VD of the superficial and deep capillary plexuses have been observed, together with inner retinal layer thinning [44]. The enlarged area of the deep FAZ continues to increase in eyes with retinal detachment and choroidal detachment, suggesting that choroidal lesions may have an acute pathological effect on ischemia of the deep retinal capillary network. Therefore, early intervention on retinal and choroidal ischemia may improve the structural recovery of the retina and the visual prognosis. Moreover, the extent of the deep FAZ may be used to predict postoperative visual acuity [45]. On the other hand, some authors have also found no correlation between FAZ, VD, and visual prognosis. The finding that eyes treated with vitrectomy have a lower VD than eyes treated with scleral buckling suggests that vitrectomy may potentially damage the microvascular structure of the vessels [46]. However, further studies are needed.

The timing of intervention for the treatment of retinal detachment is also a debated issue. High-risk indicators may facilitate the identification of eyes that benefit more than others from urgent surgery [62]. The evidence that a disrupted intraretinal appearance of the detached macula may be associated with a worse visual prognosis may aid in the selection of patients who may potentially benefit from early surgery [23]. The integrity of the photoreceptor layer seems to improve postoperatively over time. A longer time period before surgery is associated with a worse status of the photoreceptors, and it is subsequently associated with a worse visual outcome. This highlights the importance of early intervention, even in macula-off retinal detachment [24].

Studies on AO-OCT, though conducted on small samples, have highlighted that retinal reattachment after surgery does not correspond to a restoration of the outer segments of the photoreceptors, which remain considerably misaligned even after a quite long follow-up time. Structural damage to the photoreceptors prevents normal coupling of light, causing distorted and attenuated signals [4].

Postoperative photoreceptor integrity may represent a predictor of better postoperative functional outcomes. Discontinuity of the outer retinal layers and development of outer retinal folds, which are also associated with vertical metamorphopsia if close to the fovea, seem to be associated with worse visual outcomes. The surgical technique used to treat the retinal detachment may play a role in the genesis of these alterations, as suggested by some recently published studies [38,39]. However, no large RCTs are available and there is still limited data on the topic, insufficient to suggest a preferred surgical strategy. Further research on the subject should be encouraged in order to identify those OCT findings that may be relevant and crucial for choosing the most appropriate surgical technique.

It has been suggested that an abnormal macular status in the postoperative period may lead to a poor visual outcome [40]. Morphological OCT changes in the macular region seem to affect both macula-on and macula-off detachments [42]. Further studies are needed in order to clarify the role of ILM peeling, especially regarding the formation of epiretinal membranes and proliferative vitreoretinopathy in the postoperative period [51]. At the same time, studies performed with en face OCT for the detection of postoperative epiretinal membranes concluded that they are usually not severe and they have only a marginal impact on postoperative visual acuity [50].

OCT findings are useful to choose among the different endotamponades available since it is known that silicone oil, but not gases, is associated with retinal thinning. The inner retinal layers are affected: mainly the ganglion cell layer and the inner plexiform layer. Their thickness may be a predictive factor to assess the final visual acuity [34]. The parafoveal inner retinal thickness may return to normal after silicone oil removal, while the peripapillary nerve fiber layer thinning remains constant. A mechanical effect caused by the pressure of silicone oil on the retina, especially in the prone position, may be assumed [35]. The mechanism of action and the effects on visual outcome are still unclear, therefore it may be advisable to use silicone oil only in complicated cases. Moreover, monitoring of retinal thinning using SD-OCT may support the decision to remove silicone oil with the correct timing in order to minimize the potential thinning effect on the retina [32,33]. Extended use of heavy silicone oil is associated with an increased risk of ERM formation [36]. It is likely that the retina is more susceptible to damage from silicone oil than the choroid since silicone oil makes contact only with the retina. OCT-A may show a reduced VD of the superficial capillary plexus only, although no alterations of FAZ and VD may be observed [35,46].

## 5. Conclusions

In conclusion, OCT provides a more detailed knowledge of the retinal structure and of its alterations. At present, there is no gold standard for the treatment of rhegmatogenous retinal detachment, and the treatment strategy needs to be tailored to each patient. Detailed information on the preoperative retina status is useful in order to improve diagnostic accuracy and identify the prognosis. Some OCT patterns may be defined as biomarkers, predictive of anatomical and functional prognoses. OCT may be also useful in defining the retinal alterations subsequent to different endotamponades, although the evidence is still limited. In this narrative review, we have taken into consideration all of the OCT techniques available at present, since each of them may contribute to increasing the knowledge and understanding of the disease. The limit of the studies analyzed is that they are mostly retrospective in nature. Differences in patients and retinal detachment characteristics are also present, as well as disparities in surgical techniques. At present, a detailed meta-analysis on the subject cannot be conducted and there is no strong evidence supporting the choice of a surgical technique based on OCT findings. Larger prospective studies should be encouraged.

## Figures and Tables

**Table 1 jcm-11-05819-t001:** Spectral-domain and swept-source OCT.

Authors	Year	Study	No. of Eyes
Lai et al.	2010	Retrospective	37
Stopa et al.	2011	Prospective	25
Dell’Omo et al.	2012	Retrospective	33
Huang et al.	2013	Retrospective	58
Nagpal et al.	2014	Prospective	30
Terauchi et al.	2015	Retrospective	49
Srydar et al.	2015	Case report	2
Tee et al.	2016	Retrospective	61
Purtskhvanidze et al.	2017	Retrospective	20
Yang et al.	2018	Case report	1
Raczynska et al.	2018	Prospective	57
Poulsen et al.	2019	Prospective	84
Noda et al.	2019	Retrospective	42
Borowicz et al.	2019	Prospective	62
Yeo et al.	2020	Retrospective	114
Mané et al.	2021	Retrospective	85
Ozsaygili et al.	2021	Retrospective	86
Felfeli et al.	2021	Retrospective	406
Klaas et al.	2021	Retrospective	102
Kumar et al.	2021	Case control	39
Muni et al.	2021	RCT	150
Zgolli et al.	2021	Prospective	90
Uemura et al.	2021	Retrospective	11
Guan et al.	2021	Retrospective	49
Baudin et al.	2021	Prospective	115
Iwase et al.	2021	Retrospective	69
Chatziralli et al.	2021	Prospective	86
Gharbiya et al.	2012	Retrospective	35
Bansal et al.	2021	Prospective	15
Hostovsky et al.	2021	Retrospective	44
Lee et al.	2021	Retrospective	30
Lee et al.	2022	RCT	83
Horozoglu et al.	2022	Retrospective	20

RCT: randomized clinical trial.

**Table 2 jcm-11-05819-t002:** OCT angiography.

Authors	Year	Study	No. of Eyes
Hong et al.	2020	Retrospective	31
Chatziralli et al.	2020	Prospective	103
Xu et al.	2020	Retrospective	71
Roohipoor et al.	2020	Prospective	45
Nam et al.	2021	Retrospective	34
Lee et al.	2021	Retrospective	30

**Table 3 jcm-11-05819-t003:** En face OCT.

Authors	Year	Study	No. of Eyes
Fukuyama et al.	2019	Retrospective	33
Comet et al.	2021	Case report	2
Matoba et al.	2021	Retrospective	64

**Table 4 jcm-11-05819-t004:** Three-dimensional OCT.

Authors	Year	Study	No. of Eyes
Hisatomi et al.	2018	Retrospective	68

**Table 5 jcm-11-05819-t005:** Adaptive optics OCT.

Authors	Year	Study	No. of eyes
Reumueller et al.	2020	Prospective	5

**Table 6 jcm-11-05819-t006:** Potential biomarkers.

Type of OCT	Potential Biomarkers
SD-OCT and SS-OCT	Integrity of ellipsoid zone
	Integrity of external limiting membrane
	Thickness of the outer retinal layers
	Central retinal thickness
	Thickness and integrity of the inner and outer segments of the photoreceptors
	Outer retinal folds and undulations
	Integrity of the detached macula
	Macular edema
	Epiretinal membranes
	Hyper-reflective foci in the ellipsoid zone
	Height of subretinal fluid
	Macular hole
	Postoperative foveal contour
	Thickness of the ganglion cells layer
	Thickness of the outer nuclear layer
	Thickness of the inner plexiform layer
	Thickness of the peripapillary nerve fiber layer
	Persistence of subfoveolar fluid
OCT-A	Subfoveal choriocapillaris vessel density
	Dimensions of the foveal avascular zone
	Vessel density of the superficial and capillary plexuses
	Choroidal flow
En face OCT	Outer retinal folds
	Epiretinal membranes
Adaptive optics OCT	Cone morphology

OCT: optical coherence tomography; SD-OCT: spectral-domain OCT; SS-OCT: swept-source OCT; OCT-A: OCT angiography.

## Data Availability

The data presented in in this study are openly available in PubMed.

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
