# Peer review of "Optical Coherence Tomography Findings in Rhegmatogenous Retinal Detachment: A Systematic Review"

_jcm, 2022, doi:10.3390/jcm11195819_

Round 1

Reviewer 1 Report

The authors covered an interesting and clinically relevant topic on the various OCT findings in rhegmatogenous retinal detachment. As the authors stated in their limitations, there are limited data available and no RCTs on this topic for a meaningful meta-analysis. Without any quantitative statistical analysis and given the entire manuscript is based on narrative review and descriptive outcome of other studies, this study is considered a narrative review (and based on limited retrospective studies). As such, the authors should avoid making potentially biased comments, based on the limited and conflicting studies, that may bias readers in selecting surgical techniques for the management of retinal detachment. For example, lines 367-372, were based on two post-hoc analysis that showed bias towards pneumatic retinopexy. In addition, lines 174-177, making comments between pars plana vitrectomy and scleral buckling for retinal detachment is not the aim of this study.

Lines 203-212: Macular hole formation and spontaneous closure as well as epithelial proliferation seems to be out of place and only briefly touched. This can be an entire topic of discussion by itself. Please consider revision. The discussion around iLM and ERM should be revised to match theme of this manuscript.

I would recommend major revision with a focus on OCT findings in retinal detachment, highlighting potential biomarkers for visual and surgical outcome – which is a very interesting topic.

Author Response

Thank you very much for your valuable comments. We have modified the manuscript following your suggestions. We have clearly stated that the manuscript is a narrative review and that no conclusion regarding the most appropriate surgical technique can be drawn. We modified every sentence which could have been interpreted as biased towards any of the available surgical techniques. We have revised the section on postoperative hole formation. Further, we only mentioned that other studies are focused on postoperative complications, and that they are not the subject of the present review. We have also included Table 6, which summarizes current OCT biomarkers.

Reviewer 2 Report

Dear Authors,

on behalf of “The Journal of Clinical Medicine” I carefully evaluated the manuscript “Optical coherence tomography findings in rhegmatogenous retinal detachment. A systematic review.”

You performed a review on OCT findings in rhegmatogenous retinal detachment. The topic is very interesting, the manuscript is well written and easy to understand. The following addition should be considered: It would be helpful to add a table and summarize the different biomarkers, to get a better overview.

Page 7, line 271: please correct: ….is not adequate to identify and….”

We are looking forward to receiving your revised paper.

Best regards, your reviewer

Author Response

Thank you very much for your valuable scomments. We have made the suggested changes and included a table (Table 6) which summarizes all the OCT biomarkers identified by analyzed studies.

Round 2

Reviewer 1 Report

Great update on this review manuscript. Table 6 is very helpful to put all findings together. I believe, based on the title of your paper, that the purpose of your review is OCT biomarkers and relevant OCT findings in RRD, not selection of surgical technique based on OCT. In fact, pre-op OCT findings (at least based on current evidence) does not influence choice of surgical technique but rather prognostic markers for visual outcome. I would recommend the authors remove surgical technique selection based on OCT given the lack of evidence. Otherwise a great review of OCT in RRD.

Author Response

Thank you for your valuable comments. We have modified and removed the parts discussing the surgical technique selection.